# Stronger Association between High Intensity Physical Activity and Cardiometabolic Health with Improved Assessment of the Full Intensity Range Using Accelerometry

**DOI:** 10.3390/s20041118

**Published:** 2020-02-18

**Authors:** Jonatan Fridolfsson, Mats Börjesson, Elin Ekblom-Bak, Örjan Ekblom, Daniel Arvidsson

**Affiliations:** 1Center for Health and Performance, Department of Food and Nutrition, and Sport Science, University of Gothenburg, SE-405 30 Gothenburg, Sweden; mats.brjesson@telia.com (M.B.); daniel.arvidsson@gu.se (D.A.); 2Institute of Neuroscience and Psychology, University of Gothenburg, SE-405 30 Gothenburg, Sweden; 3Department of Medicine and Geriatrics, Sahlgrenska University Hospital/Östra, SE-416 50 Gothenburg, Sweden; 4Åstrand Laboratory of Work Physiology, The Swedish School of Sport and Health Sciences, SE-114 86 Stockholm, Sweden; elin.ekblombak@gih.se (E.E.-B.); orjan.ekblom@gih.se (Ö.E.)

**Keywords:** frequency filtering, vigorous physical activity, ActiGraph, multivariate analysis, partial least squares regression, cardiovascular disease, SCAPIS, LIV

## Abstract

An improved method of physical activity accelerometer data processing, involving a wider frequency filter than the most commonly used ActiGraph filter, has been shown to better capture variations in physical activity intensity in a lab setting. The aim of the study was to investigate how this improved measure of physical activity affected the relationship with markers of cardiometabolic health. Accelerometer data and markers of cardiometabolic health from 725 adults from two samples, LIV 2013 and SCAPIS pilot, were analyzed. The accelerometer data was processed using both the original ActiGraph method with a low-pass cut-off at 1.6 Hz and the improved method with a low-pass cut-off at 10 Hz. The relationship between the physical activity intensity spectrum and a cardiometabolic health composite score was investigated using partial least squares regression. The strongest association between physical activity and cardiometabolic health was shifted towards higher intensities with the 10 Hz output compared to the ActiGraph method. In addition, the total explained variance was higher with the improved method. The 10 Hz output enables correctly measuring and interpreting high intensity physical activity and shows that physical activity at this intensity is stronger related to cardiometabolic health compared to the most commonly used ActiGraph method.

## 1. Introduction

As objective measurement of physical activity (PA) have become widespread in epidemiological and clinical studies, a lot of work has been put into the development of these methods. In general, the methods consist of the following steps: collection of raw data, processing to useful metrics, calibration to represent PA and reduction to output variables to be used in further analysis [1]. The most common way of doing this is to capture raw data from an accelerometer, process this data to ActiGraph (AG) counts, calibrate these counts to energy expenditure and reduce this to the number of minutes at moderate-to-vigorous intensity (MVPA). Initially, the method development was focused on the calibration of AG counts towards energy expenditure. This lead to numerous calibrations for different populations, but also different calibrations for the same populations [2]. Usually, calibration is performed by identifying cut-points of the accelerometer output representing light (LPA), moderate (MPA), vigorous (VPA) and very vigorous (VVPA) physical activity. These were set to energy expenditure levels corresponding to 1.5, 3, 6 and 9 metabolic equivalents (METs). Accelerometer output below 1.5 METs represents sedentary time (SED). Instead of improving the methods of objective measurement of PA, the development of these many calibrations obstructs comparability between studies [3].

A closer look at the calibration of AG counts towards energy expenditure reveals that the relationship reaches a plateau at higher PA intensities than walking [4]. At even higher running speeds the relationship is reversed, implying that the measured PA intensity will be lower with increased running speed. Information about the processing of the AG counts was previously not available. However, in 2017 Brønd et al. published the processing specifications of AG counts enabling investigation and improvement of raw data processing [5]. One key step of the processing of AG counts is the application of a frequency filter with a low-pass cut-off at 1.6 Hz, which implies that parts of the acceleration signal above this frequency is attenuated. Investigation of the processing of AG counts shows that the narrow frequency filter used removes the influence of higher step frequency [6]. High intensity PA is most often associated with higher step frequency, which explains the AG counts inability to assess high intensity PA. By widening the frequency filter used in the processing of raw accelerometer data to a low-pass cut-off at 10 Hz, the positive relationship between accelerometer data and PA intensity remains at high intensity PA [6]. The 10 Hz frequency cut-off is sufficient to capture the influence of step frequency as well as all acceleration related to human PA and lead to less variation between subjects performing the same activity. This suggests that the wider filter better captures variations in PA intensity and the movement pattern.

Although a 10 Hz filter better captures PA in a lab setting, a major concern when implementing a wider frequency filter in the processing of free living data is the possible side-effect of capturing noise. However, when comparing the output from the AG filter and the 10 Hz filter, epoch by epoch, the wider filter does not seem to capture more noise in a free-living setting. With this comparison it is also apparent that, relative to the 10 Hz filter, the AG output yields much more high intensity PA. This can be explained by the lab results showing that the relationship between AG output and energy expenditure is very weak at high intensity, suggesting that the epochs are mainly classified by random [7].

A main reason of PA assessment in research and clinical practice is its relation to mortality and cardiometabolic health [1,8]. Ultimately, each methodology has to be assessed by its relation to clinical variables and outcomes. Since the output from the different methods described above is highly different with regard to high intensity PA, it is necessary to assess the validity of the methods. This could be done by investigating the relationship between the different outputs and markers for cardiometabolic health (referred to as predictive validity). As the AG output does not seem to be able to capture high intensity PA accurately, no previous studies have been able to investigate the relationship between high intensity PA and cardiometabolic health properly.

The most common intensity classification using cut-points is highly dependent on exactly where the actual cut-points are set [3]. In addition, this division is a very crude way of reducing the PA intensity spectrum, removing much of the information in the accelerometer data. Dividing the intensity spectrum into many small blocks avoids the problem with cut-points while also keeping more of the information [9]. However, because of the collinear nature of the PA intensity spectrum variables a multivariate analysis approach must be applied [10].

Thus, the aim of this study was to investigate how the improved measure of physical activity intensity, using a 10 Hz frequency filter instead of the standard 1.6 Hz AG filter, affected the relationship with markers of cardiometabolic health. Our hypothesis was that the association between high intensity PA and cardiometabolic health would be stronger with the 10 Hz filter compared to the AG filter and that the total explained variation of the model would be higher.

## 2. Materials and Methods

### 2.1. Study Sample

Study participants consisted of adults from two different cohorts, the LIV 2013 study (Swedish phrase “Livsstil, Prestation, Hälsa”, “Lifestyle, Performance, Health” in English) [11] and the SCAPIS pilot study (Swedish CArdioPulmonary BioImage Study) [12]. The LIV 2013 study consisted of a random sample of the Swedish population between ages 20 and 65. The SCAPIS pilot study consisted of a random sample, stratified for socioeconomic status, from the city of Gothenburg between ages 50 and 64. More detailed description of the recruitment process and data collection are available elsewhere [11,12,13]. Ethical approval was obtained from the Regional Ethical Review Board in Stockholm for the LIV 2013 study (no. 1338-31) and from the ethics board at Umeå University for the SCAPIS study (no. 2010-228-31M). Informed consent was retrieved from all participants. The two samples were combined in order to achieve a stable statistical model.

### 2.2. Markers of Cardiometabolic Health

Six different markers for cardiometabolic health was measured; systolic blood pressure, triglycerides, total cholesterol to high density lipoprotein ratio, insulin resistance from a homeostatic model assessment (HOMA), waist to height ratio and cardiorespiratory fitness. Systolic blood pressure (mmHg) was measured by an automatic device (Omron M10-IT, Omron Health care Co, Kyoto, Japan) in the SCAPIS pilot study and by manual auscultation with a sphygmomanometer in the LIV 2013 study. A fasting blood sample was used for measuring triglycerides (mmol/L), total cholesterol to high density lipoprotein ratio and HOMA. HOMA was estimated from the fasting blood glucose (mmol/L) and insulin (pmol/L) measurements (glucose∙insulin/22.5) [9]. Cardiorespiratory fitness (mL/kg/min) was estimated from a submaximal cycle test [14]. Remaining variables were measured using standard clinical procedures [11,13]. The six variables were combined to a composite score (CS) as done in previous studies [9,15]. Each variable was standardized to a mean of zero and standard deviation of one. Before standardization all variables except fitness was reversed (turned to negative values), which implies that for all variables a higher value indicates better cardiometabolic health. The CS was calculated as the mean of the six standardized variables for each participant.

### 2.3. Physical Activity

To measure PA, raw accelerometer data was captured using ActiGraph model GT3X or GT3X+ (ActiGraph, Pensacola, FL, USA). The two models used, GT3X and GT3X+, have been shown to have high agreement and can be used interchangeably within the same study [16]. Participants were instructed to wear the accelerometer on their right hip for seven consecutive days and to remove it during sleep. The accelerometers were set to record acceleration with a sample rate of 30 Hz and a range of +/- 6 g, where 1 g is equivalent to the gravity on earth. Raw acceleration data was extracted according to the available specifications [17].

Extracted raw acceleration was processed to the output mean mg of 3 s epochs using either the standard AG filter or a modified filter with a 10 Hz low-pass cut-off. Technical details of the processing has been published previously [7]. Night time between 00:00 and 06:00 was removed from the analysis. Non-wear time was defined as at least 60 min of consecutive zeros with an allowance of up to 2 min of output between 0 and the sedentary threshold (19.1 mg (g∙10^−3^); accelerometer output equivalent to 1.5 METs, see below) [18]. This is common procedure when analyzing AG output, therefore non-wear classification of the AG output was the reference for the 10 Hz output as well. A valid measurement was considered at least four valid days, which in turn was defined as at least eight hours of wear-time [9]. Previously published energy expenditure cut-points for 1.5, 3, 6 and 9 METs representing LPA, MPA, VPA and VVPA intensity was considered reference for PA intensity [7]. The AG cut-points were 19.1, 63.0, 171.0 and 300.2 mg and the 10 Hz cut-points were 38.9, 167.2, 582.3 and 994.1 mg. To allow for more detailed analyses of the PA intensity spectrum, the intensity spectrum was divided into smaller bins. The width of the bins was chosen to ensure enough detail at low intensity in relation to the met cut-points without being too detailed on the high intensities. For the AG output the bin edges were 0, 10, 20, 40, 60 mg and so forth, increasing with 20 mg. Since the output from the 10 Hz filter is approximately four times higher than AG at high intensities the bin edges for the 10 Hz filter was 0, 40, 80, 160, 240 mg and so forth, increasing with 80 mg.

### 2.4. Sample Characteristics

The number of participants with a valid PA measurement and a complete measurement of markers for cardiometabolic health was 725. The sample characteristics are presented in Table 1. Although the age range was wider in the LIV 2013 study, the mean age of the two samples were relatively similar (51.9 years in LIV 2013 and 57.3 years in SCAPIS Pilot). This is because the number of young participants in the LIV 2013 study was small.

### 2.5. Statistical Analyses

The statistical analysis was based on a method first implemented into PA research by Aadland et al. [9]. Since dividing the PA intensity spectrum into many small bins causes high collinearity between variables, using a multiple linear regression to investigate the relationship between PA intensity and CS is not appropriate. Instead, using a multivariate analysis approach by applying a partial least squares regression (PLS) handles this problem [10]. The PLS method decomposes the PA intensity variables (bins) into one or more latent variables called PLS components, similar to a principal component analysis (PCA). However, instead of maximizing the explained variance in the original variables on each latent variable as done with PCA, PLS maximizes the components covariance with the response variable (in this case the CS) [19]. To facilitate interpretation of the model, the predictive variance of the PLS components were combined to a single predictive component by target projection (representing all the PA intensity spectrum variables) [20]. A selectivity ratio was calculated as the ratio between predicted variance and residual variance for each PA intensity variable on the target-projected component. The selectivity ratio is a variance independent metric of the association between each PA intensity variable and the response variable CS. Since variance dependent metrics could overestimate the importance of PA intensity variables with large variance but small correlation with CS, the selectivity ratio is a more appropriate way of presenting the results [20]. The direction of the selectivity ratio, positive or negative, was retrieved from the target projection loadings. In theory, the same number of PLS components as the number of input variables can be generated. However, this would overfit the model. Instead, Monte Carlo resampling was applied, randomly using half of the samples to generate a PLS model and calculate the root mean squared error. This was repeated 100 times each for an increasing number of PLS components. The number of components that generated the lowest median root mean squared error was selected in the final model [21]. The results from the Monte Carlo resampling were also used to calculate 95% confidence intervals (CI) for the selectivity ratios and the total explained variance (R^2^) of each model.

All variables were standardized before input to the PLS model. The number of PA intensity variables (bins) included was based on where there was no more association between the accelerometer output and CS. Selectivity ratio plots were used to display the association between the PA intensity spectrum variables and the CS. Data processing and statistical analyses was performed in MATLAB R2018b (MathWorks, Natick, MA, USA).

## 3. Results

In both models, there was a relationship between the PA output and CS using up to 22 PA intensity variables (bins) with the widths specified above. The proportion of participants with at least one epoch (3 s) is reported for each PA intensity bin in Figure 1. With the AG output the proportion of participants with at least one epoch never reached below 97% for any intensity bin. With the 10 Hz output on the other hand, the proportion of participants with at least one epoch started to decline already at the mid VPA level, reached 50% at the lower part of VVPA and only 7% had one or more epochs at the last intensity bin.

The association between the PA output processed using the AG filter and the CS is presented as a selectivity ratio plot in Figure 2. Sedentary time was associated with lower CS. All PA were associated with higher CS with an increasing association from the mid MPA range and upwards, peaking in the mid VPA range. At VVPA intensity the strength of the association decreased to be very weak at the highest intensity. One PLS component was used in the model and the total explained variance (R^2^) was 12.6% (CI: 12.1-13.1%).

The association between the PA output processed using the 10 Hz filter and the CS is presented in Figure 3. Like the AG output, sedentary time was associated with lower CS and all PA was associated with higher CS. The association was slightly increasing with higher intensity up to the mid VPA range where the association strengthened rapidly to peak at the VPA-VVPA cut-point. The association with CS remained for majority of the VVPA bins before declining to no association with the last bin investigated. One PLS component was used in the model and the total explained variance (R^2^) was 14.2% (CI: 13.7–14.7%).

## 4. Discussion

The main result of the present study was that when applying the new method with the 10 Hz filter, which is wider than the most commonly used AG filter, to the processing of raw acceleration data the association between PA and cardiometabolic health was shifted towards higher intensities. Although both models show a peak of the relationship at an intensity higher than regular walking, the peak of the AG output was equivalent to the walking–running transition whereas the peak of the 10 Hz output was at higher running speeds (Figure 2 and Figure 3) [7]. More importantly, the association did not decline immediately with higher running speeds with the 10 Hz output as with the AG output. Clinically, this means that higher intensities of PA were more strongly related to cardiometabolic health. These findings are of importance for future studies on the relationship between high intensity activity and health and clinically for public health messages.

With the 10 Hz output the proportion of participants with at least one epoch above the VVPA cut-point dropped rapidly (Figure 1). At the 10 Hz outputs selectivity ratio peak, 71% had at least one epoch, which was just before where the slope in Figure 1 was steepest. This suggests that the reason why the association between the 10 Hz output and CS was weakened might be due to very few participants performing PA at the intensity of the highest part of VVPA. With the AG output on the other hand, the proportion of participants with at least one epoch was very high even at the highest intensity. In this case, the reason why the association declined was not because of too few participants having data on the high intensities but that the information in the data was not related to the health outcome. Which implies that the specificity of the AG output was low. 

The total explained variation of the 10 Hz model was significantly higher than the AG model from a statistical point of view, although the absolute improvement was not large. A majority of the PA among adults was performed at lower intensities than where the difference between the 10 Hz and the AG method was obvious. Since time spent at these lower intensities also seemed to be associated with health, the majority of the information regarding the association between an individual’s PA and health was sufficiently captured with the AG output. With a larger variation in high intensity PA in the study sample, the difference between the models might have been larger. In the current study the average time spent at VPA or VVPA was 5.5 min per day with the AG output and 2.0 min per day with the 10 Hz output (Table 1), which is similar to previous studies [22,23]. Even though most people did not perform PA at the intensity where the AG output was no longer correct, the interpretation of the AG results was problematic. This should be considered when interpreting previous research based on the AG output. 

The results of the current study are in line with previous published results regarding the relationship between AG and the 10 Hz output. Lab based results show that there is no association between high AG output and energy expenditure whereas with the 10 Hz filter the association remains at high intensity [6]. Previous results also suggests that AG heavily overestimates the amount of high intensity PA compared to 10 Hz [7]. Since lab based results suggests that the 10 Hz filter better captures high intensity PA, high intensity 10 Hz output should be considered of higher quality and closer to the true PA than the AG output at the same intensity. It might seem contradictive that despite being better at capturing high intensity PA, the 10 Hz method captures less of it. However, since the AG output is calibrated towards energy expenditure even though there is no association, epochs at this intensity will be classified by random [7]. This explains the low specificity of the AG output and why the association between the AG output and CS decreased with intensity in the VVPA range (Figure 2) although most participants had data at that intensity (Figure 1). Previous results also suggest that the 10 Hz filter is not more sensitive to capturing noise than the AG filter [7]. The current study supports this finding by showing that the association between CS and sedentary and low intensity PA was similar with the different processing methods.

Similar to the results from the 10 Hz output that VVPA is strongly related to health, previous intervention studies have shown high intensity interval training (HIIT) to be very effective in improving cardiometabolic health [24]. The current results suggest that the AG output is not able to capture HIIT and therefore misses the health benefits of this kind of PA in epidemiological research. The 2018 American Physical Activity Guidelines Advisory Committee Scientific Report states that the evidence of the effect of different PA intensities on cardiometabolic health is limited [25]. The use of AG filter in the processing of raw acceleration data have obstructed studying of the effect of PA intensity but using the 10 Hz filter overcomes this problem. The report also states that there is a need for studying the long term effects of HIIT on health outcomes [25]. Applying the 10 Hz filter in epidemiological research enables studying of HIIT equivalent PA intensity without the need of intervention studies. Aside from the relation to cardiometabolic health, the impact of the 10 Hz vs AG filter should in further studies be investigated in regard to skeletal health. For example, the bone mineral density of the femoral cortical surface has been related to MVPA using AG data [26]. The optimal intensity of PA is most likely obscured by the AG filters, and proper advice therefore cannot be produced. The results suggest that the effects of high intensity PA could be underestimated in research based on AG data.

Estimation of the relationship between PA and cardiometabolic health from a cross sectional sample has several limitations, which is generally the case in physical activity research. Anthropometrics and aerobic fitness is considered relatively stable markers of cardiometabolic health but the blood sample markers and systolic blood pressure is varying on a day-to-day basis [27,28]. Physical activity also displays large variation depending on the intensity level investigated. The habitual sedentary time can be assessed reasonably well by collecting 2–3 days of data whereas 182 days of data might be necessary to capture an individual’s habitual level of VPA [29]. Other parameters in the processing of PA data such as valid day criteria and non-wear time algorithm could have a small effect on the relationship between PA and cardiometabolic health [30], but the comparison between the filtering methods is likely not affected. The cross sectional study design also limits investigation of the causation of the relationship between PA and cardiometabolic health. The strength of this study is the quality of the PA and cardiometabolic measurements performed, including a wear time of 14.1 h per day and 7.6 valid days on average (Table 1).

## 5. Conclusions

When applying a 10 Hz frequency filter in the processing of raw accelerometer data instead of the most commonly used ActiGraph (AG) filter, the strongest association between cardiometabolic health and physical activity (PA) is shifted towards higher PA intensities. The total explained variation from a multivariate analysis between the PA output and cardiometabolic health is also statistically higher with the 10 Hz filter than with the AG filter. Although most participants have AG output at VVPA intensity, the association of this intensity to cardiometabolic health is very weak. The association between 10 Hz output and cardiometabolic health is higher at the VVPA intensity level compared to AG and the reason that the association weakens at even higher intensities seems to be that there are very few subjects with data at those intensities. The 10 Hz output enables correctly measuring and interpreting high intensity PA and enables showing that PA at this intensity is stronger related to cardiometabolic health. This should be considered in the development of PA guidelines for public health as well as in clinical settings for individual PA prescriptions.

## Figures and Tables

**Figure 1 sensors-20-01118-f001:**
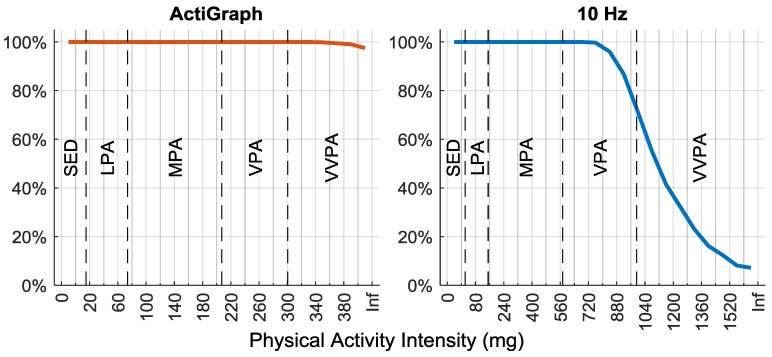
The proportion of participants with at least one epoch (3 s) for all physical activity intensity variables (bins) investigated from the ActiGraph output (left) and the 10 Hz output (right). Traditional intensity classification as reference; SED, sedentary; LPA, light physical activity; MPA, moderate physical activity; VPA, vigorous physical activity; VVPA, very vigorous physical activity; Inf, infinity.

**Figure 2 sensors-20-01118-f002:**
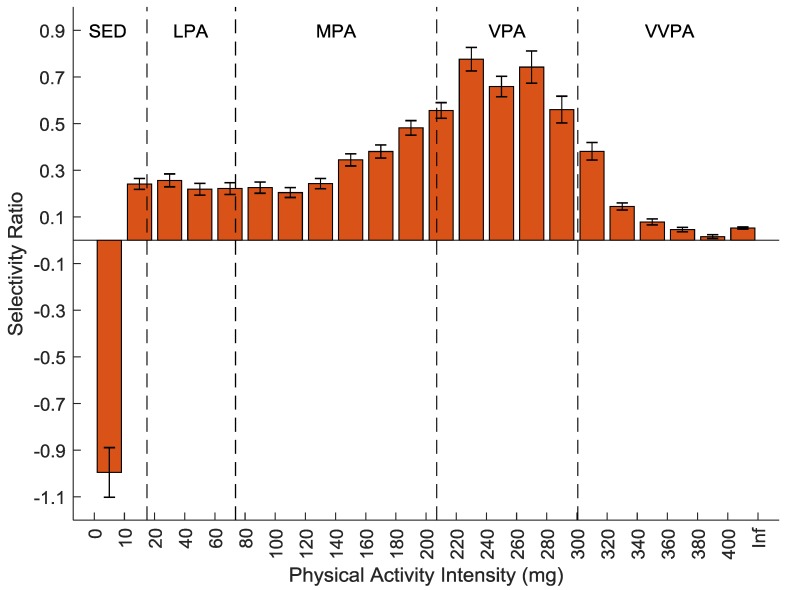
Selectivity ratio plot with confidence intervals of the association between each physical activity intensity variable (bin) using the ActiGraph filter and the cardiometabolic health composite store (CS). A positive bar indicates more time spent at that intensity is associated with higher CS and a negative bar indicates more time spent at that intensity is associated with lower CS. Traditional intensity classification as reference; SED, sedentary; LPA, light physical activity; MPA, moderate physical activity; VPA, vigorous physical activity; VVPA, very vigorous physical activity; Inf, infinity.

**Figure 3 sensors-20-01118-f003:**
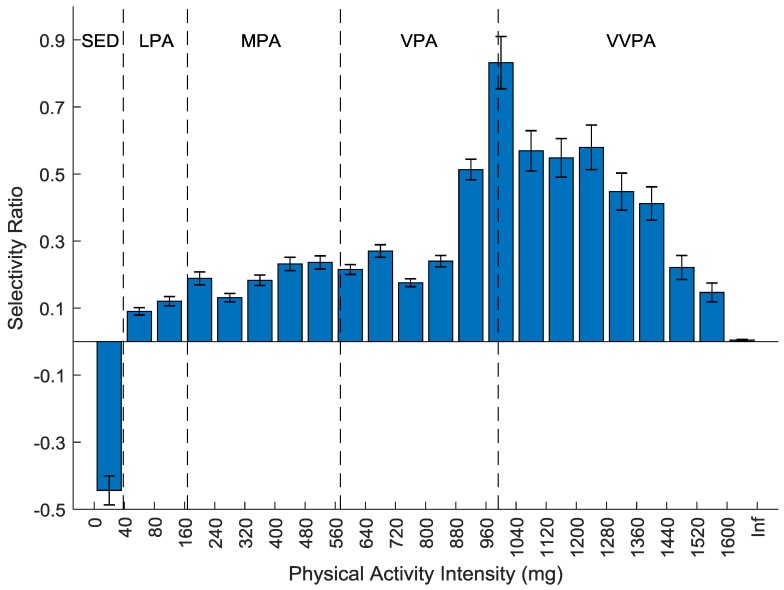
Selectivity ratio plot with confidence intervals of the association between each physical activity intensity variable (bin) using the 10 Hz filter and the cardiometabolic health composite store (CS). A positive bar indicates more time spent at that intensity is associated with higher CS and a negative bar indicates more time spent at that intensity is associated with lower CS. Traditional intensity classification as reference; SED, sedentary; LPA, light physical activity; MPA, moderate physical activity; VPA, vigorous physical activity; VVPA, very vigorous physical activity; Inf, infinity.

**Table 1 sensors-20-01118-t001:** Sample characteristics.

	Mean (SD)
N (% female)	725 (52%)
Age (years)	55.9 (7.2)
Markers of cardiometabolic health	
Systolic blood pressure (mmHg)	123.2 (16.6)
Triglycerides (mmol/L)	1.20 (0.79)
Total cholesterol:HDL (ratio)	3.59 (1.16)
HOMA (index)	2.11 (2.75)
Waist:height (ratio)	0.54 (0.06)
Fitness (mL/kg/min)	33.5 (7.4)
Physical activity assessment	
Valid days (days)	7.63 (2.13)
Wear time (hours per day)	14.1 (1.3)
Physical activity ActiGraph filter (Minutes per day)	
Sedentary	869.0 (55.3)
Light physical activity	114.9 (33.4)
Moderate physical activity	90.0 (29.5)
Vigorous physical activity	4.29 (5.29)
Very vigorous physical activity	1.17 (2.58)
Physical activity 10 Hz filter (Minutes per day)	
Sedentary	870.6 (56.0)
Light physical activity	135.0 (40.3)
Moderate physical activity	71.9 (25.6)
Vigorous physical activity	1.62 (3.46)
Very vigorous physical activity	0.37 (1.51)

^1^ SD, standard deviation; HDL, high density lipoprotein; HOMA, homeostatic model assessment.

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
