# Peer review of "Stronger Association between High Intensity Physical Activity and Cardiometabolic Health with Improved Assessment of the Full Intensity Range Using Accelerometry"

_sensors, 2020, doi:10.3390/s20041118_

Round 1

Reviewer 1 Report

Fridolfsson and coauthors provide a comparison of the relationship between PA intensity scored using a standard 1.6 Hz filter and a 10 Hz filter. Previously, the 10 Hz filter was shown in the lab to be better at picking up VPA and VVPA than the 1.6 Hz. Here, they extend these results to cardiovascular risk score. They found the 10 Hz filter produced stronger associations with the cardiovascular at higher intensities and had a greater amount of explained variance than the 1.6 Hz filter. The extent to which the differences were clinically relevant is likely not large; however, the authors posit that the mixed results of some prior studies investigating the relationship between PA intensity and cardiovascular risk may be attributable to the use of the 1.6 Hz filter.

Overall, I think this is an interesting analysis. I have a few small questions.

Questions:

Is the root cause of the 10 Hz filter overperforming the 1.6 Hz filter simply better classification? Naively, I would expect actual human movement patterns to look like the 10 Hz panel in Figure 1, which leads me to suspect that the AG 1.6 Hz filter is incorrectly classifying epochs at VPA and VVPA that are not. It was unclear in the paper why the 10 Hz filter was selected over, say, an 11 or 9 Hz filter. It would be interesting to see the change in model performance (perhaps with R^2) against the Hz used in the filter.

Comments:

The Introduction section is somewhat long. It gives a sense that the lab work showing the performance of the 10 Hz filter is definitive and does not make a strong argument for the novelty or importance of this analysis. As I understand it, the novelty and importance here comes from incorporating the cardiovascular risk scores. That could be more clear.

Typographic:

Line 88 – 1,6 Hz should be 1.6 Hz.

Line 108 – should not be indented

Table 1 has N/Age repeated at the bottom

Reviewer 2 Report

General Comments

This paper examined the differences between ActiGraph counts and an improved method with 10 Hz in a sample of two cohort studies. The authors used cross-sectional data to evaluate the levels of physical activity in relation to a cardiometabolic measure based on six different markers. They demonstrated that stronger linkages between physical activity and a cardiometabolic measure were found for higher intensity with the 10 Hz output than with the ActiGraph output. This research provided some interesting findings and may contribute to advancing the field of physical activity. However, the authors need to clarify several issues, as described below. These major and minor issues should be addressed before publication. 

Major Compulsory Comments

It was not clear why the authors used two different cohorts of adults who varied by age and socio-economic status, and so forth without providing the total sample size for each cohort. The authors should run these analyses by each cohort separately. Also, Table 1 should be stratified by each cohort, by adding more demographic characteristics, such as age, gender, SES, education, marital status, etc. The authors should perform a sensitivity test for non-wear time with the one they used, compared to the other one introduced by Choi et al. (2011) below. The definition, “Non-wear time was…sedentary threshold[18].” is unclear. The authors should clearly define it.

Reference

Choi, L., Liu, Z., Matthews, C.E., Buchowski, M.S., 2011. Validation of accelerometer wear and nonwear time classification algorithm. Med Sci Sports Exerc 43:357-64.

Why did the authors use eight hours of wear-time for a valid day? Shouldn’t  this be nine hours, instead?  Reference #18 used a sample of children. The present study used two cohorts of adults.  Reference #18 needs to be changed to a study using a sample of adults with nine,  hours as a valid day, not eight.  If the authors prefer to use eight hours, then they need to do the sensitivity analysis with nine hours as a valid day for monitoring. How important is it to define more accurate high intensity physical activity in relation to cardiometabolic outcomes? Generally speaking, not many individuals engage in vigorous intensity physical activity for a long duration. Why and how does this study improve the current physical activity research? The authors should add the limitations section in the discussion section. The authors should define mg greater in detail. Table 1 should be moved to the results section, rather than being before statistical analyses.

Minor Compulsory Comments

Lines 48-50, what is the confusion in the field? In Table 1, why did the authors list a row for N (%female) and Age (years) twice? This does not make sense.  
